# Prognostic and Predictive Factors in Advanced Head and Neck Squamous Cell Carcinoma

**DOI:** 10.3390/ijms22094981

**Published:** 2021-05-07

**Authors:** Teresa Magnes, Sandro Wagner, Dominik Kiem, Lukas Weiss, Gabriel Rinnerthaler, Richard Greil, Thomas Melchardt

**Affiliations:** 1Oncologic Center, Department of Internal Medicine III with Haematology, Medical Oncology, Haemostaseology, Infectiology and Rheumatology, Paracelsus Medical University, 5020 Salzburg, Austria; t.magnes@salk.at (T.M.); sa.wagner@salk.at (S.W.); d.kiem@salk.at (D.K.); lu.weiss@salk.at (L.W.); g.rinnerthaler@salk.at (G.R.); r.greil@salk.at (R.G.); 2Cancer Cluster Salzburg, 5020 Salzburg, Austria; 3Salzburg Cancer Research Institute-Laboratory for Immunological and Molecular Cancer Research (SCRI-LIMCR), 5020 Salzburg, Austria

**Keywords:** head and neck squamous cell carcinoma, biomarker, human papilloma virus, checkpoint inhibitors, combined positive score, tumor mutational burden, tumor microenvironment

## Abstract

Head and neck squamous cell carcinoma (HNSCC) is a heterogeneous disease arising from the mucosa of the upper aerodigestive tract. Despite multimodality treatments approximately half of all patients with locally advanced disease relapse and the prognosis of patients with recurrent or metastatic HNSCC is dismal. The introduction of checkpoint inhibitors improved the treatment options for these patients and pembrolizumab alone or in combination with a platinum and fluorouracil is now the standard of care for first-line therapy. However, approximately only one third of unselected patients respond to this combination and the response rate to checkpoint inhibitors alone is even lower. This shows that there is an urgent need to improve prognostication and prediction of treatment benefits in patients with HNSCC. In this review, we summarize the most relevant risk factors in the field and discuss their roles and limitations. The human papilloma virus (HPV) status for patients with oropharyngeal cancer and the combined positive score are the only biomarkers consistently used in clinical routine. Other factors, such as the tumor mutational burden and the immune microenvironment have been highly studied and are promising but need validation in prospective trials.

## 1. Introduction

Head and neck squamous cell carcinoma (HNSCC) is a heterogeneous disease that develops from the mucosal tissue of the oral cavity, the pharynx, and the larynx and accounts for more than 650,000 new cancer diagnoses and 330,000 deaths globally per year [1]. From a worldwide perspective, tobacco use and alcohol consumption are the most common risk factors and the combined misuse of both substances potentiates the risk for developing HNSCC [2,3]. In some regions of Asia betel nut chewing was identified as an independent risk factor for developing oral and oropharyngeal cancer [4]. Furthermore, infection with human papillomavirus (HPV) mainly contributes to the rising numbers of oropharyngeal squamous cell carcinoma (OPSCC) in the United States (US) and Western Europe [5].

The overall outcome of patients with locally advanced HNSCC is poor and even after modern, multimodality treatments between 40–50% of the patients experience disease recurrence [6,7]. Until recently, the standard of care (SOC) for patients with recurrent or metastatic HNSCC, incurable by concurrent chemoradiation and surgery, was cetuximab, an anti-epidermal growth factor receptor (EGFR) antibody, in combination with a platinum agent and fluorouracil. The overall response rate (ORR) to this regimen was 36% in the original study published by Vermorken et al. in 2008 and the median progression free survival (PFS) was 5.6 months [8]. Today, the two programmed cell death protein 1 (PD-1) inhibitors nivolumab and pembrolizumab are approved for patients with recurrent or metastatic HNSCC and pembrolizumab alone or in combination with platinum and fluorouracil is the SOC for the first-line treatment in patients with PD-L1 positive tumors. However, in platinum-resistant disease, the response rates to PD-1 inhibition lie between 13–18% and some studies even show a higher early mortality for checkpoint inhibitors compared to active control arms [9,10,11,12]. In the KEYNOTE-048 study, the PFS-rate at 12 months was 17% in the overall population of patients who were treated with pembrolizumab alone or in combination with chemotherapy [13]. This suggests that only a small proportion of patients benefit from immunotherapy.

Overall, these studies show that there is an urgent need to improve the treatment strategies for patients with advanced HNSCC. Apart from testing new drugs and treatment combinations, this can be achieved by investigating factors that help to improve prognostication and treatment selection. In this review, we describe the currently most relevant prognostic and predictive factors in HNSCC and discuss their roles and limitations.

## 2. HPV Infection and p16

Over the last two decades, there was a decrease in smoking- related HNSCC in the US and some countries in Europe. However, the numbers of OPSCC cases in the US, Denmark, and Scotland began to rise, and this was explained by the increase of HPV-associated cancers at this anatomic site [14,15]. The link between HPV 16 infection and the development of squamous cell carcinomas at the tonsillar region and the base of the tongue was shown in two large meta-analyses of case-control studies [16,17]. Other high-risk HPV genotypes, such as HPV 18, 31, and 33 are also associated with OPSCC but are found less frequently. The viral DNA of high-risk HPV types consists of several genes that lead to the expression of early (E1-7) and late (L1-2) proteins in the host cell. The two oncogenes E6 and E7 are mainly responsible for the cancerous transformation in the mucosal cells. E6 leads to the degradation of p53 and E7 causes the binding and destabilization of retinoblastoma (Rb), another tumor suppressor [18]. Recent studies have shown that in the western world around 70–80% of OPSCC cases are caused by HPV infection, which makes it the most common HPV-associated cancer in these regions [19,20]. HPV infection is less common in squamous cell carcinomas of the head and neck outside the oropharynx with a prevalence between 11–24%, depending on the exact site [21,22].

HPV positive OPSCC is nowadays seen as a distinct entity when compared to HPV negative disease due to the differences in the molecular profiles (see Table 1), clinical presentation, and prognosis. This has led to the development of a new staging system for HPV positive OPSCC that was presented in the most recent, eighth edition of the tumor, node, metastasis (TNM) classification of the Union for International Cancer Control (UICC)/American Joint Committee on Cancer (AJCC) [23]. The staging system was based on the results of the ICON-S trial, which showed that patients with locally advanced HPV positive OPSCC and multiple ipsi- and contralateral cervical lymph node metastases have a favorable prognosis that does not differ from patients with none or only one lymph node metastasis [19]. In the study the HPV status was either determined by in situ hybridization (ISH) or p16 immunohistochemical staining (IHC), a surrogate parameter for HPV infection that is also used for the new TNM classification. The tumor suppressor P16INK4A (p16) regulates the cell cycle by binding to the cyclin D1 CDK4/CDK6 complex and thereby blocking the phosphorylation of the Rb protein. The HPV protein E7 inactivates Rb and consequently causes the overexpression of p16 in infected cells [24]. Several other trials have also shown that p16 positivity is associated with a significantly better prognosis in patients with OPSCC compared to patients with p16 negative disease [25,26,27]. As an example, in the trial that led to the approval of definitive radiotherapy in combination with cetuximab, patients in the experimental arm with p16 positive OPSCC had a 3-year overall survival (OS) rate of 87.8% compared to 41.9% in patients with p16 negative tumors [27]. However, it is now known that p16 positivity does not always correlate with HPV positivity identified by other detection methods such as HPV DNA polymerase chain reaction (PCR), DNA ISH, and RNA ISH [21,28,29]. In a meta-analysis including 25 studies and more than 5000 patients p16 IHC and HPV PCR/ISH showed consistent results in 86.0% of the patients (35.6% HPV positive/p16 positive and 50.4% HPV negative/p16 negative) but 6.7% of the patients were HPV negative/p16 positive and 7.3% were HPV positive/p16 negative. Interestingly, only the double positive group showed a significantly better OS compared to all other patients, while the HPV negative/p16 positive group had an intermediate outcome, and the two other groups had the worst prognosis [28]. Similar results were seen in the recently presented EPIC-OPC study which included around 7700 patients from 13 international cohorts. In this study 11% of the patients would have had a wrong classification by p16 IHC alone. Again, the patients that were both HPV positive and p16 positive had the best outcome but in this trial both the HPV positive/p16 negative and the HPV negative/p16 positive groups showed intermediate results concerning the 5-year disease free survival (DFS) and OS and the double negative patients had the worst outcomes [30]. One reason for the inconsistent results of p16 IHC and HPV PCR/ISH might be false negative results of one of the tests and indeed, one research group described that by adding mRNA ISH as a third test for patients with positive p16 IHC and negative HPV DNA ISH, a group of patients could be classified as truly HPV positive with similar performance as those that were p16 positive/HPV DNA ISH positive [31]. However, false testing alone would not explain the distinct clinical behavior of tumors with discordant test results. It has been suggested that HPV PCR and/or ISH positive but p16 negative tumors might not be directly driven by the virus infection and that HPV is merely an innocent bystander in this subgroup [28]. This testing issue could be overcome by measuring virus activity through detection of mRNA or proteins of HPV oncogenes. The exact mechanism of p16 overexpression in HPV negative disease is currently unknown but it could also be linked to alterations of Rb and the cyclin D1 pathway [32,33]. Overall, these data show that p16 testing alone can lead to an understanding of a significant group of patients. Currently, the American Society of Clinical Oncology (ASCO) recommends p16 IHC for the determination of the HPV status in all patients with OPSCC and additional test methods only at the discretion of the pathologist and the treating physician [34]. Furthermore, the most recent staging system does not include the smoking history, despite data showing that it influences the prognosis of patients with p16 positive OPSCC [35].

P16 expression and HPV infection are also prognostic in patients with OPSCC and local and/or distant recurrence. A combined analysis of two studies with locally advanced disease reported the data of 181 patients with known p16 status and locoregional only (55%) and distant or combined distant and local (45%) relapse of disease [39]. In this cohort, the median OS for patients with p16 expression was 2.6 years compared to 0.8 years for patients with p16 negative tumors (HR 0.49, 95% CI 0.39–0.70). Similar results were seen when other tests for the detection of HPV were carried out in addition to p16 IHC [40,41].

The exact prognostic role of HPV infection and p16 positivity in non-oropharyngeal HNSCC is still unclear and at present, routine testing is not recommended [34]. However, recent analyses have also shown an association between the detection of high-risk HPV and improved OS in hypopharynx and larynx cancer [22,42,43]. Tian et al. also reported a favorable prognosis in patients with locally advanced oral cavity cancer and HPV positive disease, while others saw no link between HPV infection and OS at this anatomic site and even detected worse DFS in patients with HPV positive oral cavity cancer [22,43].

Some studies have also suggested a predictive role of p16 expression for the treatment with chemoradiotherapy or EGFR-inhibitors [44,45,46]. In the SPECTRUM and the LUX-H&N1 trial p16 positivity was associated with reduced benefit from the addition of panitumumab to chemotherapy and afatinib compared to methotrexate [45,46]. However, in the EXTREME study, where p16 IHC was combined with HPV detection using Cervista^®^ HPV 16/18 and Cervista^®^ HPV HR assays, there was no difference in the treatment effect between p16 and/or HPV positive and negative patients [41].

The higher rate of CD4+ cells, the larger infiltration of activated CD8+ T cells, and the lower count of regulatory T cells in the microenvironment of HPV positive OPSCC shows that these tumors are less immunosuppressive when compared to HPV negative disease [47,48]. This has led to the hypothesis that HPV positivity is predictive of a better response to checkpoint inhibitors in HNSCC. Indeed, some trials reported higher response rates in patients with p16 and/or HPV positive tumors. In the KEYNOTE-012 study, which investigated pembrolizumab in pretreated, recurrent/metastatic HNSCC the overall response rate of patients with p16 positive disease was 24% compared to 16% in patients with p16 negative tumors [49]. Similar results were seen with the use of durvalumab and nivolumab in this setting [12,50]. Patients with HPV positive disease analyzed by p16 IHC, HPV ISH or PCR had a response rate of 29.4% to durvalumab compared to 10.8% in patients with HPV negative tumors and 15.9% of p16 positive patients responded to nivolumab compared to 8% of p16 negative patients. However, in the KEYNOTE-040 and −055 studies, these differences in response were not seen, and in the clinically most relevant KEYNOTE-048 study, benefit from pembrolizumab was seen independently from p16 status [13,51,52].

The favorable prognosis of patients with HPV positive OPSCC has led to several completed and ongoing studies with the goal to de-intensify treatment for this population. Recently, the ECOG-ACRIN 3311 study showed that transoral resection is a safe and effective approach in patients with p16 positive OPSCC and that patients with low-risk disease might need less intensive postoperative therapy after transoral resection [53]. However, de-intensification through the use of radiotherapy combined with cetuximab instead of radiotherapy with cisplatin was not successful according to the latest results of two other trials [54,55]. Until today, there is therefore no role for treatment de-intensification for HPV positive OPSCC outside a clinical trial. Furthermore, it is important to note that only patients with HPV positive OPSCC and stages I-II (T1-3, N0-N2) and a smoking history below 10 pack-years are considered to have a truly “low-risk” disease and should be included in de-escalation trials [56].

## 3. Tobacco Use

Smoking tobacco products remains to be one of the major risk factors for developing HNSCC and there is a clear increase of risk with a higher frequency and longer duration of smoking habits [3,57,58]. Different studies reported that between 26–56% of patients are current smokers at the time of diagnosis of HNSCC and the smoking rates are highest in patients with larynx cancer [59,60,61,62]. This is important to note as the smoking history, apart from tumor stage and location, is one of the most relevant prognostic clinical factors in patients with HNSCC. Current or former tobacco use is not only associated with a higher rate of comorbidities and second primary cancers but also leads to a higher disease-specific mortality and several studies have shown that the risk of death is approximately twice as high in current or former smokers compared to non-smokers [59,61,63]. Furthermore, it has been known for a long time that patients with HNSCC who continue to smoke during radiotherapy have lower response rates and shorter survival compared to patients who quit smoking before treatment [64,65].

As mentioned above, several reports have also suggested that smoking alters the course of disease in patients with HPV positive OPSCC and described that tobacco users with HPV positive tumors have a higher rate of recurrence and distant metastases and worse prognosis compared to never-tobacco users with HPV positive disease [66,67]. In an analysis of two large phase III trials including patients with OPSCC one pack year and one year of smoking increased the risk of progression or death by 1% and 2%, respectively, after adjusting for other risk factors including p16 [68]. In one of these trials patients with OPSCC could be separated into three groups with significantly different outcomes according to HPV status, smoking history, and stage of disease and these results were later validated in a large single-center database [35,69].

Tobacco use has a strong immunosuppressive effect on the tumor microenvironment (TME) in patients with HNSCC and therefore also seems to influence the response to checkpoint inhibition in patients with HNSCC [70]. In the CHECKMATE-141 study that led to the approval of nivolumab in patients with platinum-resistant disease the treatment effect of the anti-PD-1 inhibitor was smaller in the subgroup of smokers compared to non-smokers. Another retrospective analysis also showed that the outcome of patients with current or former tobacco use treated with checkpoint inhibitors was worse compared to never smokers. Importantly, this was only true for patients with HPV negative disease and there was no statistically significant difference between smokers and non-smokers with HPV positive tumors [71].

## 4. PD-L1 Status and Combined Positive Score

Checkpoint inhibitors are antibodies that target cytotoxic T lymphocyte Antigen 4 (CTLA-4), PD-1, and programmed death ligand-1 (PD-L1) and thereby block the inhibition of the immune response by the tumor and its microenvironment. These drugs have improved the outcomes of patients with numerous different tumor types and have the potential to create long-term survival in patients with metastatic disease [72,73,74,75]. The results of some of the phase-I trials suggested that a higher PD-L1 expression in tumor cells or tumor cells and immune cells combined correlates with a higher response rate to checkpoint inhibition [76,77]. This made PD-L1 the most important and widely used predictive biomarker for immunotherapy and various PD-L1 cut-offs have been used as inclusion criteria or stratification factors in past and ongoing studies [13,78,79].

Most of the studies investigating the prognostic value of PD-L1 expression in patients with HNSCC treated with surgery, radiation, chemoradiotherapy or combined modalities show that high expression is associated with a worsened outcome [80,81]. The results of two independent cohorts including patients with local disease from different anatomical areas suggested that PD-L1 expression is a strong negative prognostic factor for OS, independent from other known risk factors, such as tumor stage, resection status, and extracapsular lymph node expansion [82]. Interestingly, in a large meta-analysis there was no clear association between PD-L1 expression alone and OS but patients with low levels of tumor infiltrating lymphocytes (TILs) and high PD-L1 expression combined also had worse prognosis [83].

The predictive value of PD-L1 expression in tumor cells or tumor cells and immune cells combined has been evaluated in all trials investigating checkpoint inhibitors in HNSCC. However, different cut-offs and PD-L1 expression locations were used, which complicates the interpretation of the study results [11,12,13,51,84] (see Table 2).

The KEYNOTE-040 and -048 studies investigating pembrolizumab in advanced HNSCC both used the combined positive score (CPS), which is the number of PD-L1 positive cells including tumor cells, lymphocytes, and macrophages divided by all viable tumor cells and multiplied by 100 [13,51]. In the KEYNOTE-040 study patients with platinum resistant recurrent or metastatic HNSCC were randomized between pembrolizumab and investigator’s choice (IC; methotrexate, docetaxel or cetuximab) and there was a statistically significant survival benefit for pembrolizumab in the overall population. However, subgroup analysis showed that this survival advantage was only significant in patients with a CPS ≥ 1 (OS 8.7 months vs. 7.1 months, *p* = 0.0078) and patients with a CPS < 1 had no clear benefit from pembrolizumab compared to IC. Furthermore, the KEYNOTE-040 trial also showed that patients with a tumor proportion score (TPS) ≥ 50%, indicating the percentage of tumor cells with membranous PD-L1 expression, had a significantly improved overall response and survival outcome with pembrolizumab [51]. The CHECKMATE-141 study used similar inclusion criteria as the KEYNOTE-040 study and investigated nivolumab versus IC. There was a significant survival benefit for nivolumab, which lead to its approval by the U.S. Food and Drug Administration (FDA) and European Medicines Agency (EMA). In this study PD-L1 expression was only evaluated on tumor cells and the thresholds of ≥1%, ≥5%, and ≥10% showed no clear correlation with improved survival outcomes [12]. Interestingly, longer follow-up led to an increased benefit of nivolumab compared to SOC in patients with PD-L1 negative disease [85]. In the KEYNOTE-048 study treatment naïve patients with recurrent or metastatic HNSCC were randomized between pembrolizumab alone, pembrolizumab in combination with platinum and fluorouracil, and the EXTREME regimen (carboplatin or cisplatin, fluorouracil and cetuximab). The combination of pembrolizumab and chemotherapy improved survival compared to the SOC in patients with PD-L1 expressing tumors (CPS ≥ 20 and CPS ≥ 1) as well as in the total population. Pembrolizumab alone was superior in patients with a CPS ≥ 20 and in those with a CPS ≥ 1 and was noninferior in the overall population while showing a favorable safety profile [13]. In 2019 these results led to the EMA approval of the checkpoint inhibitor pembrolizumab alone or in combination with platinum and fluorouracil in the first line setting of patients with HNSCC and a CPS ≥ 1%. The FDA approved the combination regimen for all patients but pembrolizumab as a single agent was also restricted to patients with PD-L1 positive disease. Therefore, the determination of the CPS in patients with advanced HNSCC is now crucial for treatment selection. The EAGLE-trial was another randomized phase III trial investigating checkpoint inhibition in patients with advanced HNSCC. In this study durvalumab and durvalumab plus tremilimumab were compared to SOC in patients with platinum-resistant disease and there was no benefit for checkpoint inhibition. The PD-L1 status was determined on tumor cells only and cut-offs of ≥25% and ≥1% were used. However, there was also no benefit for durvalumab alone or in combination with tremilimumab in PD-L1 positive patients. In the overall population, the investigators even noted a higher early mortality sign in patients treated with checkpoint inhibition compared to SOC and saw a crossing of the survival curves with longer follow-up [11]. Overall, these studies suggest that patients with PD-L1 positive HNSCC benefit more from checkpoint inhibitors than patients with PD-L1 negative disease and that looking at the PD-L1 expression in tumor and immune cells combined might be more predictive. However, the study results are heterogeneous and a proportion of PD-L1 negative patients might also benefit from immunotherapy [85]. Apart from using different assays and cut-offs, the heterogeneity of the study results could be explained by other factors such as varying PD-L1 expression between the primary and metastatic tumor site [88].

## 5. Next Generation Sequencing and Tumor Mutational Burden

The broad implantation of next generation sequencing (NGS) techniques provides a detailed insight into the mutational landscape of various tumor entities. The analysis of 74 HNSCC samples by Stransky et al. showed that their mutational profile is similar to that of other smoking-associated malignancies like small cell lung cancer and lung adenocarcinoma. Apart from genes known to be recurrently mutated in HNSCC such as *TP53*, *CDKN2A*, *PETN* or *PIK3CA*, alterations were also enriched in genes that control squamous cell differentiation (e.g., *NOTCH1*, *IRF6*, and *TP63*) [89]. While the overall mutational load of HPV-negative and HPV-positive HNSCC seems to be similar, the specific genes involved differ largely between the two subgroups (see Table 1) [37,38,90]. Mutations in *TP53* and *CDKN2A* are detected almost exclusively in HPV negative disease because the HPV proteins E6 and E7 lead to the elimination of p53 and Rb [37]. *MLL2/3*, *CUL3*, *NSD1*, *NOTCH1*, and *PIK3CA* mutations are also common in HPV-negative tumors whereas unique loss of *TRAF3* and amplification of *E2F1* and frequent aberrations in *KRAS, PIK3CA, FAT1, FGFR2/3,* and *NOTCH1* were reported in HPV-positive disease [37,38]. Furthermore, the presence of specific alterations seems to influence the course of disease in HNSCC as mutations in *TP53*, *NOTCH1*, and *CDKN2A* were associated with inferior survival outcomes and patients responding to cisplatin were more likely to harbor *FAT1* mutations [90]. The analysis of paired samples from patients with HNSCC at primary diagnosis and distant recurrence by massively parallel sequencing showed a complex pattern of evolution, similarly to that seen in other tumor entities [91,92].

NGS analyses also allow the determination of the tumor mutational burden (TMB), which is usually defined as the number of somatic mutations per DNA megabase (Mb). A high TMB is thought to provide a high number of neoantigens and therefore a higher immunogenicity of the tumor, resulting in improved response to checkpoint inhibition. Indeed, a landmark analysis of 27 tumor types by Yarchoan et al. showed that TMB is significantly correlated with the ORR to anti-PD-1 or anti-PD-L1 therapy [93]. Further reports of retrospective and prospective studies confirmed that high TMB is associated with improved response and survival rates in tumors such as non-small cell lung cancer (NSCLC), urothelial cancer, and melanoma [94,95,96]. In patients with HNSCC treated with pembrolizumab the TMB was also predictive of improved response, independent of other factors such as PD-L1 expression and T-cell inflamed gene expression profile (GEP) [97,98,99]. The results on TMB and response to checkpoint inhibition in patients with HPV-positive tumors are conflicting as one analysis showed that TMB correlates with improved overall response, independent of the HPV status while the results of another study only reported improved outcomes in patients with high TMB and HPV negative disease [97,100]. In the latter analysis by Hanna et al. the tumors of 126 HNSCC patients treated with anti-PD-1/anti-PD-L1 therapy at a single center were evaluated by massively parallel sequencing and immune profiling. Patients with HPV-negative tumors and a TMB greater than 10 had significantly improved overall survival compared to those with a TMB below 5 (median OS 20.0 vs. 6.0, *p* = 0.01) but there was no difference in OS in patients with HPV-positive disease. In the same study smokers had a significantly higher TMB, independent of the viral status but smoking could not predict the outcome in the multivariate analysis [100]. In the above-mentioned EAGLE-trial the TMB was evaluated in plasma samples before treatment. Patients with a blood TMB (bTMB) ≥ 16mut/Mb had significantly improved OS with durvalumab or durvalumab plus tremilimumab compared to chemotherapy while there was no clear benefit from checkpoint inhibition in patients with low bTMB (hazard ratios (HR) for OS for durvalumab vs. chemotherapy in patients with bTMB ≥ 16 0.39 (95% confidence interval (CI) 0.20–0.75) and in patients with bTMB < 16 HR 0.91 (95% CI 0.61–1.37)). The bTMB was independent of other clinical and prognostic factors such as HPV status, PD-L1 expression, age, gender, tumor location, and ECOG performance score [101]. Overall, these studies indicate that high TMB predicts improved benefit from checkpoint inhibition in HNSCC but the lack of a consistent threshold and correlation with other biomarkers such as PD-L1 expression preclude its use as a single biomarker. Interestingly, a study of patients with locally advanced HNSCC treated with combined chemoradiation showed that high TMB was correlated with poor outcome suggesting that this group might benefit from treatment with checkpoint inhibition [102].

A recent report showed that HNSCC samples with pathogenetic mutations of the tumor suppressors *TP53* and/or *CDKN2A* had a higher mean TMB compared to samples with wildtype genes [103]. Similar results were seen in a study by Klinakis et al. in which *TP53* mutations were also associated with a higher TMB score in the metastatic tissue of patients with HNSCC but not in the primary tissue. Importantly, this study also showed that despite the higher TMB, patients with metastatic disease harboring *TP53* mutations had worse survival undergoing treatment with checkpoint inhibitors compared to patients with wildtype *TP53* [104]. This shows that *TP53* might be another important biomarker in patients with HNSCC and should be determined when TMB is evaluated.

Over the last years a growing number of research articles was published in the field of transcriptomics and epigenomics in HNSCC, with the majority focusing on prognostic signatures [105]. Some studies also developed gene expression models that could be used to predict treatment response. For example, the radiosensitivity index (RSI) based on the expression of ten hub genes was correlated with improved 2-year locoregional control in 92 patients with locally advanced HNSCC treated with concomitant radiochemotherapy [106]. The RSI was then used to develop a genome-based model for adjusting radiotherapy dose based on the results from patients with 20 different tumor entities [107]. Furthermore, a 15-gene hypoxia classifier defined patients with more hypoxic tumors that were associated with a worse clinical outcome after radiotherapy. This negative prognostic effect was overcome by treatment with the hypoxic modifier nimorazole [108]. A recent study tested 16 gene expression signatures of radiosensitivity, HPV status, tumor hypoxia and microsatellite instability together with established clinico-pathological risk factors in patients with OPSCC. Several signatures for HPV and microsatellite instability were significantly correlated with OS. However, the authors also reported high inter-tumor heterogeneity and the need to improve the quality control and validation of gene expression studies to further improve current risk stratification for patients with OPSCC [109].

## 6. Tumor Infiltrating Lymphocytes

Apart from cancer-associated fibroblasts and components of the extracellular matrix the TME of HNSCC also consists of a number of different immune cells such as TILs (CD8+ T-cells and regulatory T-cells (Tregs)), macrophages, natural killer cells, and myeloid derived suppressor cells [110]. In general, the TME of patients with HNSCC is in an immunosuppressive state. This is caused by various mechanisms such as reduced HLA class I expression and secretion of immunosuppressive cytokines of tumor cells, T-cell anergy of peripheral T-cells and TILs and overexpression of PD-1 and CTLA-4 [111]. However, the TME of HPV positive and HPV negative tumors differs, and HPV positive tumors show a higher rate of CD8+ TILs compared to HPV negative tumors [47,112]. A higher level of CD8+ TILs is associated with improved survival and better response to checkpoint inhibitors in several different tumor types [113,114,115]. Similar findings were seen in both HPV positive and HPV negative HNSCC where a higher number of CD8+ TILs predicts improved DFS and OS [116]. The generally increased immunogenicity of HPV positive tumors might also explain their improved prognosis as one study showed that the outcome of patients with virus associated tumors with low TILs was similar to those of patients with HPV negative tumors while survival was significantly improved for patients with HPV positive tumors and high TILs [117]. The Immunoscore (IS) is a measurement of CD3+ and CD8+ infiltrates in the tumor center and the invasive margin and was first validated as a prognostic biomarker in non-metastatic colon cancer [118]. Zhang et al. showed that the IS also predicts DFS and OS in patients with HNSCC who had undergone complete resection and that it further separated the prognosis of patients with the same tumor stage [119]. Furthermore, it was reported that patients with HNSCC who respond to neoadjuvant chemotherapy are more likely to have a high IS compared to non-responders [120]. In the above-mentioned retrospective single-center analysis of patients treated with anti-PD-1/anti-PD-L1 agents by Hanna et al., the CD8+ TILs of patients not responding to therapy were more likely to co-express the inhibitory immune-checkpoint molecules TIM-3 or LAG-3 with PD-1 [100]. Apart from the evaluation of TILs, some studies also showed a predictive value of circulating immune cells, for example in patients with melanoma and NSCLC [121,122]. In the CHECKMATE-141 study a group of patients received nivolumab beyond radiologic tumor progression. A subgroup analysis showed that patients initially responding to therapy and those treated beyond progression had a significantly higher percentage of CD8+ T-cells but a lower rate of PD-1+ CD8+ effector cells and also less PD-1+ T-regs in the peripheral blood before treatment start compared to those not treated beyond progression [123]. Overall, these studies show that TILs and circulating immune cells have the potential to serve as prognostic and predictive biomarkers in HNSCC but confirmation in prospective trials is needed.

## 7. Conclusions

HNSCC is a heterogenous disease and several clinical and molecular factors may help to improve diagnosis and treatment selection. Until today the only validated and routinely used biomarkers are p16 IHC in patients with OPSCC and the CPS for treatment selection in patients with recurrent or metastatic disease. Due to the continuing efforts to de-intensify treatment for patients with HPV associated OPSCC it is important to combine p16 IHC with other HPV detection methods in order to avoid understaging and therefore undertreatment of these patients. Because of the results of the KEYNOTE-048 study and the specific approval of pembrolizumab by the FDA and EMA, the determination of the CPS is mandatory in patients with recurrent or metastatic disease. However, the results of other trials investigating checkpoint inhibitors in HNSCC were inconsistent and it is therefore still unclear which PD-L1 cut-off and detection method serve best as a biomarker in HNSCC. The reports on TMB and TILs as predictive biomarkers for immunotherapy in HNSCC are interesting but the routine determination of these biomarkers in daily routine is still challenging and they also need validation in prospective trials.

## Figures and Tables

**Table 1 ijms-22-04981-t001:** Characteristic molecular alterations in HPV positive and HPV negative HNSCC.

HPV Positive OPSCC	HPV Negative HNSCC
Oncogenic transformation caused by viral proteins E6 and E7 leading to degradation of p53 and Rb [18]	Carcinogen exposure through tobacco smoke, alcohol, betel nut chewing causes DNA damage and inaccurate DNA repair [36]
Unique and frequent loss of *TRAF3* and amplification of *E2F1* [37] Frequent aberrations in *KRAS, PIK3CA, FAT1, FGFR2/3,* and *NOTCH1* [37,38]	Unique and frequent alterations in *CDKN2A* and *TP53* [37] Frequent alterations in *MLL2/3, CUL3, NSD1, NOTCH1,* and *PIK3CA* [37,38]

HPV = human papillomavirus, OPSCC = oropharyngeal squamous cell carcinoma, HNSCC = head and neck squamous cell carcinoma.

**Table 2 ijms-22-04981-t002:** Phase III studies testing checkpoint inhibitors in recurrent/metastatic HNSCC and associated PD-L1 cutoffs.

Agent	Study	PD-L1 Expression Location	Cut-Off	PFS ^1^ (HR)	OS ^1^ (HR)	Median OS ^1^ (Months)
Nivolumab [85]	CHECKMATE-141 (nivolumab vs. SOC in platinum resistant patients)	TCs	<1 ≥1%	1.13 0.59	0.73 0.55	6.5 vs. 5.5 8.2 vs. 4.7
Pembrolizumab [51]	KEYNOTE-040 (pembrolizumab vs. SOC in platinum resistant patients)	TCs + ICs	CPS < 1 CPS ≥ 1	n.a. 1.01	1.28 0.74	6.3 vs. 7.0 8.7 vs. 7.1
		TCs	TPS < 50% TPS ≥ 50%	n.a. n.a.	0.93 0.53	6.5 vs. 7.1 11.6 vs. 6.6
Pembrolizumab [13,86,87]	KEYNOTE-048 (pembrolizumab alone or in combination with CTX in platinum sensitive patients)	TCs + ICs	CPS < 1 CPS 1–19 CPS ≥ 1 CPS ≥ 20	4.31 1.25 1.16 0.99	1.51 0.86 0.78 0.61	7.9 vs. 11.3 10.8 vs. 10.1 12.3 vs. 10.3 14.9 vs. 10.7
Durvalumab [11]	EAGLE (durvalumab or durvalumab + tremilimumab vs. SOC in platinum resistant patients)	TCs	<25% ≥25%	n.a. n.a.	n.a. n.a.	7.6 vs. 8.0 9.8 vs. 9.0

^1^ Comparison of PFS and OS resulting from nivolumab vs. methotrexate, docetaxel or cetuximab in the CHECKMATE-141 study, pembrolizumab vs. methotrexate, docetaxel or cetuximab in the KEYNOTE-040 study, pembrolizumab vs. platinum, fluorouracil and cetuximab in the KEYNOTE-048 study and durvalumab vs. cetuximab, taxane, methotrexate or fluouracil in the EAGLE trial; PFS = progression free survival, OS = overall survival; HR = hazard ratio; SOC = standard of care; TCs = tumor cells; ICs = immune cells; TPS = tumor proportion score (percentage of tumor cells with membranous PD-L1 expression); CPS = combined positive score (number of PD-L1 positive cells (tumor cells, lymphocytes, and macrophages) out of the total number of tumor cells × 100); n.a. = not available, vs. = versus.

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
