# Peer review of "Prognostic and Predictive Factors in Advanced Head and Neck Squamous Cell Carcinoma"

_ijms, 2021, doi:10.3390/ijms22094981_

Round 1

Reviewer 1 Report

In this study, the authors summarized several “biomarkers” related to head and neck squamous cell carcinoma (HNSC). Several sections were discussed, as (1) HPV infection and p16, (2) tobacco use, (3) PD-L1 status, (4) next-generation sequencing and mutational burden, (5) tumor infiltration lymphocytes. The authors concluded that all these studies need further validation by prospective trials.

The overall flow of this paper seems random and out of focus. There is no logical connection in the section topics. It is unclear what the authors intend to address, the diagnostic biomarkers, prognostic biomarkers, therapeutic prediction biomarkers, or molecular pathological biomarkers. Nevertheless, the authors described more content related to PDL-1 immunotherapy. I suggested authors re-organized this paper and address on this specific issue. 

Author Response

Dear Reviewer,

Thank you for critically discussing our manuscript. While we respect your opinion of our review, we would like to point out that the aim of the review was to summarize the biomarkers in head and neck squamous cell carcinoma (HNSCC) that are currently most discussed and most relevant for clinical routine. We outlined the different categories of biomarkers in the introduction and in the revised manuscript tried to clarify the role of the individual factors that were described. As PD-1 inhibitors have become part of the standard of care for patients with locally incurable recurrent or metastatic HNSCC a large part of the recent literature in the field has focused on biomarkers for checkpoint inhibition. We hope that you still find our manuscript suitable for publication.

Yours sincerely,

Thomas Melchardt

Reviewer 2 Report

The review "Evolving biomarkers in advanced head and neck cancer" is a very well written review  with the latest information of the field.

I have only one comment:

In the introduction the authors write " the standard of care (SOC) for patients with locally advanced HNSCC, ineligible radiotherapy and surgery". In the Nordic countries the SOC for advanced HNSCC has been chemoradiotherapy (cisplatin)  and often in combination with surgery for at least the last ten years. What about Austria?

Author Response

Dear Reviewer,

Thank you very for the positive evaluation of our manuscript. In Austria, as well as in the Nordic countries, the standard for patients with locally advanced disease eligible for surgery and radiotherapy is also combined radiation and chemotherapy with cisplatin with or without prior resection. However, in the introduction of the review we are discussing the treatment for patients with recurrent or metastatic head and neck squamous cell carcinoma incurable by concurrent chemoradiation and/or surgery. In this situation, the former standard was a combination of cisplatin or carboplatin, fluorouracil and cetuximab. We tried to clarify this section in the revised manuscript. We hope that you now our manuscript suitable for publication.

Yours sincerely,

Thomas Melchardt

Reviewer 3 Report

Overall, a well written paper although not very novel. Note minor typographical and punctuation errors. Whilst the paper is covering HNSCC as a whole, there is a lot of focus on oropharyngeal cancers as opposed to oral or laryngeal cancers - perhaps include a paragraph in each section addressing each of the other two sites so as to better reflect the diversity of HNSCC.

Areas which should be included in the review:

  • Alcohol risk factors are not discussed
  • Betel nut?
  • Table 1 - might be useful to do a meta-analysis for better visualisation of results (could compare low vs high and list cut-offs beside each study)
  • molecular biomarker section is rather limited, only looking at TMB and specific gene mutations. What about novel modalities such as transcriptomic analysis, epigenetics etc.

Author Response

Dear Reviewer,

Thank you very much for the positive evaluation of our review and the insightful comments to improve our work.

The typographical and punctuation errors were corrected in the revised manuscript.

You also mentioned that a strong focus was laid on oropharyngeal cancer and suggested to include more information on tumors from other sites. Indeed, the section on human papillomavirus (HPV) infection and p16 focuses on oropharyngeal cancer as this is most relevant for clinical routine and HPV testing is only recommended for this anatomic site. In the revised manuscript we included the recent data on the prognosis of HPV positive nonoropharyngeal cancer.

As you suggested, we discussed the risk factors for the development of HNSCC, including alcohol abuse and betel nut chewing, in more detail in the revised introduction of the manuscript.

We thank you for the suggestion to include a meta-analysis in the table of our review. However, we fear that due to the different cut-offs, PD-L1 detection methods and inclusion criteria of the studies such an analysis would be misleading.

Lastly, you suggested to include a section on transcriptomic analyses in HNSCC. We thank you for pointing out this interesting field of research. A paragraph on selected gene expression analyses was added to the molecular biomarker section of our review.

We hope that you now find our manuscript suitable for publication.

Yours sincerely,

Thomas Melchardt

Round 2

Reviewer 1 Report

In the revision, the authors do not have much changed in the organization of this paper. The most critic of this paper is that the Title does not match the Content. For example, in the first two parts, the P16 status and the condition of tobacco use, are both not the “biomarkers.” In the first part, the authors need to make a Table to summarize the molecular alteration (biomarkers) related to HPV infection or the P16 status. Similarly, in the second part, authors may make a Table or a Figure to indicate the most significant molecules (biomarkers) that may be induced by tobacco in head-neck cancer patients. So as and so on.

Author Response

Dear Reviewer,

Thank you very much for the reevaluation of our review and the helpful comments to improve our manuscript. We agree that in head and neck squamous cell carcinoma it can be difficult to clearly differentiate between classic biomarkers and risk factors for the disease. We therefore changed the title of our manuscript to “Prognostic and predictive factors in advanced head and neck squamous cell carcinoma”. As you suggested, we also included a table (Table 1 in the revised manuscript) to summarize the molecular alterations in HPV positive and HPV negative/ nicotine associated head and neck cancer.

We hope that you now find our manuscript suitable for publication.

Yours sincerely,

Thomas Melchardt

Round 3

Reviewer 1 Report

In this revision, the authors have made a significant change, by modification the title to match the content of the manuscript. There is no major concern for this paper and may be suitable for publication.